# Improved YOLOv8 for Gas-Flame State Recognition under Low-Pressure Conditions

**DOI:** 10.3390/s24196383

**Published:** 2024-10-02

**Authors:** Qingyi Sai, Jin Zhao, Degui Bi, Bo Qin, Lingshu Meng

**Affiliations:** 1College of Energy and Power Engineering, University of Shanghai for Science and Technology, Shanghai 200093, China; 2College of Environment and Architecture, University of Shanghai for Science and Technology, Shanghai 200093, China

**Keywords:** low-pressure environment, flame detection, YOLOv8 algorithm, attention mechanism

## Abstract

This paper introduces a lightweight flame detection algorithm, enhancing the accuracy and speed of gas-flame state recognition in low-pressure environments using an improved YOLOv8n model. This method effectively resolves the aforementioned problems. Firstly, GhostNet is integrated into the backbone to form the GhostConv module, reducing the model’s computational parameters. Secondly, the C2f module is improved by integrating RepGhost, forming the C2f_RepGhost module, which performs deep convolution, extends feature dimensions, and simplifies the inference structure. Additionally, the CBAM attention mechanism is added to enhance the model’s ability to capture fine-grained features of flames in both channel and spatial dimensions. The replacement of CIoU with WIoU improves the sensitivity and accuracy of the model’s regression loss. Experimental results on a simulated dataset of the theoretical testbed indicate that compared to the original model, the proposed improvements achieve good performance in low-pressure flame state detection. The model’s parameter count is reduced by 12.64%, the total floating-point operations are reduced by 12.2%, and the detection accuracy is improved by 21.2%. Although the detection frame rate slightly decreases, it still meets real-time detection requirements. The experimental results demonstrate that the feasibility and effectiveness of the proposed algorithm have been significantly improved.

## 1. Introduction

Gas furnaces, which use methane as their primary fuel, have stringent requirements for combustion stability. Unstable combustion or flameout events can easily trigger explosive accidents. Low-pressure and low-oxygen environments at high altitudes can cause significant changes in key parameters, such as methane combustion efficiency [1], flame shape [2], and flame stability [3]. These changes affect the safety of the combustion process. Therefore, precise identification of flame status is crucial for real-time monitoring and safety warnings during combustion, directly impacting life and property safety.

Early gas-flame detection technology primarily relied on hardware devices such as infrared cameras, temperature sensors, and pressure sensors [4]. However, these devices are expensive and complex to maintain. The rapid development of flame image detection technology, which offers fast and accurate detection capabilities, has been widely applied in various fields. Cai Min [5] improved the VIBE algorithm and segmented the surveillance video, enhancing detection speed and accuracy. Amin Khatami et al. [6] used K-medoid clustering and Particle Swarm Optimization (PSO) for fire detection. Compared to traditional methods, their proposed method is faster and more accurate. Haijun Zhang et al. [7] proposed a detection method using SVM-trained samples, based on hybrid linear function kernels and radial basis function kernels, to reduce false negative rates and improve flame detection accuracy.

In recent years, with the rapid development of deep learning technology, traditional image detection techniques have gradually been replaced by deep learning models, becoming the mainstream technology for flame detection. Hatice Catal Reis et al. [8] used transfer learning based on ImageNet weights and achieved high accuracy with the DenseNet121 model. Guillermo Ronquillo-Lomeli [9] used the Probabilistic Neural Network (PNN) algorithm to classify boiler flames, achieving an accuracy rate of over 90%. Hongyang Zhao et al. [10] combined YOLOv8 with VQ-VAE to enhance the accuracy of actual flame detection. To reduce the false positive rate of traditional models, Hikmat Yar [11] improved the YOLOv5s model by adding a stem module to the backbone, replacing the Neck with a smaller kernel, and adding a P6 module to the Head. This achieved significant progress in reducing model size, complexity, false positive rate, and improving accuracy. Guanbo Wang [12] proposed an improved lightweight flame detection model based on YOLOv4-tiny, increasing the receptive field and pruning the model, which significantly reduced the number of model parameters and weight files while enhancing accuracy. Lumeng Chen et al. [13] improved the YOLOv5s model by adding a coordinate attention mechanism to the feature extraction part and modifying α-CIoU, enhancing the model’s convergence speed.

Although these studies have yielded good results in flatland environments, in plateau conditions, the flame’s edge shape, area, and color change rapidly. As atmospheric pressure decreases, the expansion of the flame’s edge becomes more pronounced, and the area increases. Additionally, the flame color transitions from bright yellow to pale blue, while the decrease in oxygen concentration raises the flue gas content, complicating the recognition and feature analysis of flame images. Deepening the network to address these challenges often results in reduced or lost flame image information, which can necessitate an increase in parameters and a consequent reduction in detection speed. To overcome these issues, this paper introduces a flame state detection algorithm based on an improved YOLOv8. The algorithm optimizes the model’s backbone with a lightweight design, significantly enhancing detection speed and enabling deployment on mobile devices. Additionally, the incorporation of an attention mechanism improves the model’s ability to capture fine-grained flame features, thereby enhancing detection accuracy. Performance analysis and practical tests confirm the effectiveness and feasibility of the proposed method.

## 2. Introduction to YOLOv8

The YOLO (You Only Look Once) series is renowned for its optimal balance of speed and accuracy, rendering it ideal for mobile device deployment. In January 2023, the original v5 development team, Ultralytics, released the latest version of this series, YOLOv8, which is available in five specifications: n, s, m, l, and x. YOLOv8 inherits and improves upon the features of its predecessors, delivering outstanding performance, especially for flame image object detection.

YOLOv8’s architecture consists of three parts: backbone, Neck, and Head, as illustrated in Figure 1. The backbone part comprises Conv, C2f, and SPPF modules, designed to extend network depth and the receptive field, enabling the fusion of local and global features. The Neck part includes C2f, Concat, and Upsample modules, integrating feature maps at different scales through a path aggregation network and C2f module, promoting vertical information fusion. The Head part adopts a decoupled structure, divided into classification and localization prediction branches, helping to resolve conflicts between the two. Compared to previous versions, YOLOv8 introduces an anchor-free design, using the sigmoid function to output the probability of detected objects and determining target objects through a threshold. YOLOv8 incorporates the Task Aligned Assigner for one-stage object detection and uses Distribution Focal Loss for regression loss to improve detection accuracy. These improvements enable YOLOv8 to demonstrate exceptional object detection, tracking, and segmentation performance.

## 3. Improved YOLOv8 Algorithm

This paper proposes a lightweight YOLOv8-based flame category detection model. The improvements in this model are mainly reflected in four aspects:To address the issue of excessive model parameters, the GhostConv module from the GhostNet network is used to replace the standard Conv modules in the backbone layer.The C2f module is upgraded to C2f_RepGhost, and both GhostConv and C2f_RepGhost employ reparameterization techniques to effectively reduce computational complexity.To enhance detection accuracy, the CBAM (Convolutional Block Attention Module) attention mechanism is integrated into the Neck and Head components of the model.To improve regression performance, IoU is utilized in place of the original CIoU, thereby enhancing the model’s convergence capability. The improved architecture is illustrated in Figure 2.

### 3.1. Ghostconv Module

The Noah’s Ark Lab at Huawei designed a lightweight network architecture known as GhostNet [14], which maintains the dimensions and channel sizes of the original convolutional output feature maps while reducing the computational and parameter load of the model.

In practical applications, the input image data, $X\in R^{h\times w\times c}$, consist of c input channels, with h and w representing the height and width of the image, respectively. The output image data, $Y\in R^{h^{\prime }\times w^{\prime }\times n}$, have n output channels, with $h^{\prime }$ and $w^{\prime }$ representing the height and width. The convolution kernel size is k×k, and the kernel size for linear operations is d×d. The input image data are divided into s parts, as follows:

The parameter count for a traditional convolutional layer is
(1)n×c×k×k

The parameter count for the Ghost module is
(2)$$\frac{n}{s}\times c\times k\times k+(s-1)\times \frac{n}{s}\times d\times d$$

The compression ratio at this time is
(3)$$r_{c}=\frac{n\times c\times k\times k}{\frac{n}{s}\times c\times k\times k+(s-1)\times \frac{n}{s}\times d\times d}\approx s$$

The theoretical acceleration ratio for upgrading a standard convolution with the Ghost module is
(4)$$\begin{array}{c} r_{s}=\frac{n\times h^{\prime }\times w^{\prime }\times c\times k\times k}{\frac{n}{s}\times h^{\prime }\times w^{\prime }\times c\times k\times k+(s-1)\times \frac{n}{s}\times h^{\prime }\times w^{\prime }\times d\times d}\\ \;\;\;=\frac{c\times k\times k}{\frac{1}{s}\times c\times k\times k+\frac{s-1}{s}\times d\times d}\approx s \end{array}$$

Therefore, Ghost Convolution (GhostConv), a convolutional module in the GhostNet network, effectively enhances the model’s efficiency and performance compared to standard convolution. GhostNet initially employs a 5 × 5 convolutional kernel to extract flame feature information from the input flame feature map, as illustrated in Figure 3. These feature maps undergo inexpensive linear transformations with a stride of 1 to derive new feature maps. The final feature map is obtained by concatenating these maps, reducing computational costs while preserving relevant feature information across different flame types.

### 3.2. Improved C2f Module

The RepGhost (Representative Ghost) module is a lightweight network structure designed to reduce computational resource consumption while maintaining model performance [15].

The reparameterization process of the RepGhost module is depicted in Figure 4. It involves generating an equivalent convolutional layer in the batch normalization (BN) branch and integrating this convolutional layer with the BN layer. During training, the RepGhost module applies deep convolution to input features, expanding feature dimensions. Subsequently, batch normalization (BN) enhances the nonlinearity of the training process, with this step potentially merged during inference. The feature maps produced by both branches are then added together, preserving the same number of channels. Finally, a ReLU activation function is applied to comply with the reparameterization rules, ensuring efficient inference.

The RepGhost module significantly simplifies its structure during the inference stage by fusing features in the weight space rather than the feature space. By consolidating parameters from both branches, the RepGhost module achieves a streamlined inference structure comprising only standard convolutional layers and ReLU activation functions, thereby enhancing algorithm efficiency. In the YOLOv8 network, the C2f module merges low-level and high-level feature maps, enhancing detection accuracy and robustness. However, the C2f module introduces additional computational complexity to the model, resulting in longer training and inference times.

To address this issue and enhance the model’s efficiency, we integrated the C2f module with the RepGhost module to create a new module, C2f_RepGhost. This hybrid module preserves flame images’ multi-scale feature fusion capabilities while reducing the model’s computational load. By incorporating the RepGhost module, the C2f_RepGhost module effectively improves inference and detection speed, benefiting the deployment of flame detection models on devices with limited hardware performance. Consequently, the C2f_RepGhost module lowers the demand for hardware resources while maintaining model performance, rendering the flame detection model more practical and deployable.

### 3.3. CBAM Attention Mechanism

Flame state recognition involves a fine-grained recognition problem [16]. Incorporating an attention mechanism enables the model to swiftly identify crucial areas in flame images, thereby enhancing image classification speed. This mechanism mimics human visual perception by focusing on key features while suppressing unnecessary ones [17]. CBAM (Convolutional Block Attention Module) [18] is a lightweight convolutional attention module known for significantly boosting model performance when integrated into convolutional neural networks. CBAM introduces minimal additional parameters and computational demands, making it suitable for integration into various convolutional networks.

Flame image feature maps contain diverse information across different dimensions: the channel dimension conveys details such as flame color and thermal radiation, while the spatial dimension captures the flame’s shape, size, and texture features. The CBAM module integrates the Channel Attention Module (CAM) and the spatial attention module (SAM). These submodules operate sequentially to generate attention feature maps in both the channel and spatial dimensions. They leverage inter-channel feature relationships to produce channel attention maps, which are then combined with spatial attention to yield the most informative feature maps. The overall structure is depicted in Figure 5.

Given an intermediate feature map F as input, the channel attention map $M_{c}$ is computed first, followed by the spatial attention map $M_{s}$ from the spatial attention module. The resultant feature map $F^{\prime }$ is then obtained through a weighted operation, formulated as follows:(5)$$\left\{\begin{array}{l} F=M_{c}(F)⊗F\\ F^{\prime }=M_{s}(F^{\prime })⊗F^{\prime } \end{array}\right.$$

In the formula, $M_{c}(F)-F$ represents the output weights after channel attention, and $M_{s}(F^{\prime })-F^{\prime }$ represents the output weights after spatial attention. The “⊗“ symbol denotes the weighted multiplication operation for feature maps.

### 3.4. Loss Function

In YOLOv8, the conventional bounding box regression loss function has been replaced with CIoU Loss (Complete Intersection over Union Loss) to address the challenge of effectively distinguishing different regression cases. CIoU Loss is an advanced loss function that not only considers the overlapping area between bounding boxes (Intersection over Union, IoU) but also incorporates the distance between the centers of bounding boxes and ensures consistency in aspect ratios [19]. However, CIoU Loss still has limitations; it struggles with highly overlapping or intersecting bounding boxes and is sensitive to changes in the size and shape of bounding boxes during optimization.

To mitigate these challenges, this paper adopts WIoU, which introduces a weighting mechanism when calculating IoU to better account for relative positions, sizes, and shape differences between bounding boxes [20]. The fundamental principle of WIoU involves assigning different weights to various regions and features of bounding boxes, thereby enhancing the sensitivity and accuracy of regression loss. The mathematical expression of this approach is detailed below:(6)$$L_{WIoU}=r\exp(\frac{{\left(x-x_{gt}\right)}^{2}+{\left(y-y_{gt}\right)}^{2}}{W_{g}^{2}+H_{g}^{2}})L_{IoU}$$
(7)$$L_{IoU}=1-\frac{W_{i}H_{i}}{wh+w_{gt}h_{gt}-W_{i}H_{i}}$$
(8)$$r=\frac{\beta }{\delta \alpha^{\beta -\delta }}$$
(9)$$\beta =\frac{L_{IoU}}{\bar{L_{IoU}}}\in [0,+\infty )$$

In this context, x, y, w, and h represent the predicted bounding box’s centroid coordinates (horizontal and vertical), width, and height, respectively. Correspondingly, $x_{gt}$, $y_{gt}$, $w_{gt}$, and $h_{gt}$ represent these dimensions for the actual bounding box. $W_{g}$ and $W_{H}$ denote the width and height of the smallest enclosing box covering both the predicted and actual bounding boxes, while $W_{i}$ and $H_{i}$ are the width and height of the overlapping region between them. $\bar{L_{IoU}}$ is the moving average of $L_{IoU}$. β represents the degree of outlier presence, with higher values indicating poorer sample quality.

The WIoU regression loss function dynamically adjusts the gradient gain of anchor boxes based on outliers, aiming to maximize the gradient gain of anchor boxes during training. By using WIoU, the competitiveness of high-quality anchor boxes is reduced, and the gradients produced by low-quality examples are also diminished. This shift in focus to average-quality anchor boxes improves the overall performance of the detector.

## 4. Experimental Design and Results Analysis

### 4.1. Dataset Construction

The experimental image data were obtained from the low-pressure gas combustion mechanism test bench at the University of Shanghai for Science and Technology. Taking Lhasa, a typical high-altitude city at 3650 m altitude with an atmospheric pressure of approximately 65 kPa, as an example, the experimental setup simulated its low-pressure environment using a vacuum pump and frequency converter to maintain the pressure in the combustion chamber around 65 kPa. Figure 6 illustrates the experimental setup. Oxygen, nitrogen, and methane were introduced into the combustion chamber at different equivalent ratios. An electronic ignition device ignited the mixture, and industrial CCD cameras captured images of gas flames under low-pressure conditions, including premixed stable flames, diffusion stable flames, premixed unstable flames, and diffusion unstable flames.

The collected video data were converted into PNG images using Python. To enhance the dataset’s diversity and improve the model’s generalization capability, data augmentation operations were performed on the images using the OpenCV library. These operations included random adjustments of image brightness, saturation, mirroring, and the addition of random Gaussian noise. The results of these operations are depicted in Figure 7. In total, 8140 images were obtained, comprising 3005 images of premixed stable flames, 1015 images of premixed unstable flames, 3100 images of diffusion stable flames, and 1020 images of diffusion unstable flames. All images were annotated using LabelImg and divided into a training set and a test set in an 8:2 ratio, resulting in 6512 images for the training set and 1628 images for the test set. The components are shown in Table 1:

### 4.2. Experimental Setup and Parameter Configuration

The experiment was conducted on a 64-bit Windows 11 operating system with the following environment: PyTorch 2.0.0, CUDA 11.7, and Python 3.9.16. The hardware specifications of the computer are as follows: Intel(R) Core(TM) i5-12490F CPU, 32 GB of RAM, and an NVIDIA GeForce RTX 2080 Ti GPU with 22 GB of memory. The model was trained for 200 epochs to enhance data fitting, with a batch size of 16. The optimization was performed using the classical SGD optimizer, with an initial learning rate of 0.001 and a final learning rate of 0.01 to improve convergence stability. Additionally, Mixup was set to 0.2 and Degrees to 5 to increase data diversity, thereby enhancing the model’s generalization ability and robustness.

### 4.3. Evaluation Indicators

To intuitively demonstrate the performance improvement of the model after the enhancements, the following metrics were used to evaluate the overall performance of the model: precision (P), recall (R), mean average precision (mAP), number of parameters (Params), total floating-point operations (FLOPs), frames per second (FPS), and F1 score. Recall measures the ability to identify correct samples, precision is used to evaluate the accuracy of predictions, and recall assesses the ability to find the correct samples. The definitions of precision, recall, and F1 are as follows:(10)$$P=\frac{TP}{TP+FP}$$
(11)$$R=\frac{TP}{TP+FN}$$
(12)$$F1=\frac{2\times P\times R}{P+R}$$

In the formula, TP (true positive) refers to the number of correctly predicted positive samples; FP (false positive) refers to the number of negative samples incorrectly predicted as positive; and FN (false negative) refers to the number of positive samples incorrectly predicted as negative. The average precision (AP) is calculated by integrating the precision–recall (PR) curve. The mean average precision (mAP) is then obtained by averaging all AP values. A higher mAP value indicates better model detection performance. The formula is as follows:(13)$$AP=\int_{0}^{1}P(r)dr$$
(14)$$mAP=\frac{1}{n}\sum_{i=1}^{n}AP$$

This study evaluates the effects of the improved model by analyzing metrics such as mean average precision (mAP), the number of parameters, computational load, model size, and the detection speed of flame targets in videos.

## 5. Experimental Results

### 5.1. Data Distribution

In order to analyze the distribution of targets in the flame detection task, we initially conducted a visualization of the dataset, focusing on the categorization of targets, their spatial distribution, and the dimensions of the targets. This analysis also revealed the geometric characteristics of the targets. As illustrated in Figure 8, the majority of targets are concentrated in the central region of the image, with a significant density around the origin of the *x*-axis. The y-coordinate is concentrated around 0.5, indicating that the flames typically occupy the central region of the image. Furthermore, the width and height of the flame targets fall within the ranges of 0.075 to 0.125 and 0.2 to 0.4, respectively. This indicates that the flames display a considerable vertical extension.

### 5.2. Comparative Experiments

In the training of the enhanced model proposed in this study and the YOLOv8n model, hyperparameter settings for both models were kept consistent. The F1 scores for each model were compared throughout the iterations. Figure 9 presents the line chart depicting the F1 scores of both models over 200 training epochs. Initially, during the first 30 epochs, the F1 scores of the enhanced model were comparable to those of YOLOv8n, attributable to the absence of pre-trained weights. Between epochs 30 and 50, significant fluctuations were observed in the F1 scores of both models, with the enhanced model exhibiting higher F1 scores compared to YOLOv8n. After epoch 50, the models stabilized, with the enhanced model consistently outperforming YOLOv8n in terms of F1 scores.

Table 2 provides a comparative analysis of the enhanced model and YOLOv8n in terms of precision, recall, and mAP@50. The enhanced model achieved 21.2% higher precision, 7.4% higher recall, and 0.5% higher mAP@50 compared to YOLOv8n. These results indicate that the enhanced model surpasses YOLOv8n in detection accuracy. Additionally, following lightweight optimizations, the enhanced model demonstrated a notable reduction in the number of parameters, with total floating-point operations reduced by 12.2% and model parameters reduced by 12.64%. Thus, the enhancements have significantly contributed to the model’s lightweight nature.

To further assess the performance of the enhanced model, the Grad-CAM (Gradient-weighted Class Activation Mapping) technique was employed to compare and analyze the attention mechanisms of YOLOv8n and the proposed model. The results, depicted in Figure 10, demonstrate that incorporating the CBAM (Convolutional Block Attention Module) significantly enhances the model’s performance in detecting flames under low-pressure conditions. Predictions on the test set flame data reveal that the model excels in accurately identifying and classifying flames, exhibiting high confidence and precise localization capabilities. Representative prediction results are illustrated in Figure 11.

To verify the effectiveness of the improved module, comparative experiments were conducted with other YOLO models under the same hyperparameter settings. The experimental results are shown in Table 3. Compared with the YOLOv5m, YOLOv5l, and YOLOv7 models, the improved model has significant advantages in lightweight design, with detection accuracy increased by 3.8%, 8.5%, and 8.4%, respectively, and detection speed improved by 45 frames, 12 frames, and 56 frames, respectively. Compared with the YOLOv7-tiny and YOLOv8s models, the number of model parameters decreased by 56.4% and 15.3%, total floating-point operations decreased by 45.5% and 12.2%, detection accuracy improved by 10.5% and 9.1%, and detection frame rate increased by 47 frames and 17 frames, respectively. However, there was a slight decrease in recall, F1, and mean precision.

Overall, the improved model proposed in this paper shows significant improvements in lightweight design and detection speed compared to other models. It reduces the number of parameters and computational load without compromising detection accuracy, thereby lowering memory resource consumption. This makes it well suited for deployment in gas-flame detection devices.

### 5.3. Model Training and Comparison

Figure 12a shows the training loss for all models, which decreases rapidly in the early stages and stabilizes after about 50 rounds. The training loss curves of the improved models are very close to those of the other models, which indicates that all models perform relatively consistently on the training set, with no obvious signs of overfitting. Figure 12b shows the validation loss, which also shows a similar trend: most of the models drop rapidly in the first few rounds and then gradually stabilize, and the validation loss curves are close to those of the other models, suggesting that the models have good generalization ability.

In order to evaluate the generalization ability of the models, we contacted the relevant organizations to obtain some flame data from a boiler at an altitude of 2800 m. These data were used to test two models, the original dataset model and the augmented dataset model. The test results are shown in Figure 13, and both models show high confidence. Especially after data enhancement, the confidence of the models is further improved, indicating that our data enhancement strategy effectively enhances the model’s adaptability and generalization to new environments.

### 5.4. Ablation Study

An ablation study was conducted to assess the performance improvements of the model with various added modules. The study examined four configurations: M1, representing the GhostConv module; M2, representing the GhostConv + C2f_RepGhost module; M3, representing the GhostConv + C2f_RepGhost + CBAM module; and M4, representing the GhostConv + C2f_RepGhost + CBAM + WIoU module. The results, presented in Table 4, show that after incorporating the GhostConv module, the model’s accuracy increased by 20.1%, the F1 score improved by 6.2%, and the total floating-point operations decreased by 2.43%. This indicates that the GhostConv module contributes to the model’s lightweight design. However, there was a 7.3% decrease in recall and a 0.3% drop in mAP, suggesting that while GhostConv reduces computation, it also diminishes the model’s ability to capture feature information, resulting in missed positive samples and reduced recall and mean average precision.

Incorporating the C2f_RepGhost module increased the recall rate by 2.9%, but accuracy decreased by 1.2% and mAP dropped by 7.37% compared to the baseline model. The number of model parameters decreased by 11.5%, and the total floating-point operations were reduced by 12.2%, indicating that this module significantly reduces computation but also impairs the model’s ability to capture fine-grained features of flames, thereby affecting detection performance.

The addition of the CBAM attention mechanism led to a 21.1% increase in accuracy compared to the model with only the C2f_RepGhost module, with improvements of 3.1% in F1 score and 8.27% in mAP. This demonstrates that the CBAM module effectively enables the model to focus on and extract fine-grained features of flame images.

Finally, replacing the loss function with WIoU resulted in a 10.5% increase in overall accuracy and a 5.3% increase in mean average precision, indicating that WIoU enhances the model’s detection accuracy. Figure 14 and Figure 15 illustrate that WIoU surpasses other IoUs in terms of accuracy, mean average precision, and F1 score.

## 6. Conclusions

In this paper, we propose an improved YOLOv8n flame state detection method specifically optimized for low-pressure gas flames in plateau environments. By replacing the Conv and C2f modules in the backbone part of the original model with the GhostConv and C2f_RepGhost modules, respectively, we achieve a significant lightweighting of the model. In addition, the introduction of the CBAM attention mechanism further improves the model’s ability to capture fine-grained features of the flame, making it more efficient in recognizing flame features. The improved loss function enables the model to pay more attention to key flame features by introducing WIoU, which significantly improves the ability to distinguish the flame from the background, thus enhancing the robustness and accuracy of the model.

The experimental results show that the improved YOLOv8n model performs well in terms of lightweighting and accuracy improvement. Specifically, the amount of total floating-point operations is reduced by 12.2%, the amount of model parameters is reduced by 12.64%, the accuracy is improved by 21.2%, and the recall is improved by 7.4%. These improvements enable the model to effectively reduce the consumption of computational resources under low-pressure conditions in a plateau environment, while providing an efficient and feasible solution.

However, this study also has some limitations. First, the improved model was mainly validated in a specific low-pressure flame detection task, and its performance in other environmental conditions still needs to be further evaluated. Second, although the model has improved in accuracy and robustness, its ability to detect extreme flame features still needs to be further optimized. Future research can consider extending the applicability of the model under different environmental conditions, exploring more effective feature extraction techniques, and incorporating real-time processing requirements to further improve the overall performance and adaptability of the flame detection system.

## Figures and Tables

**Figure 1 sensors-24-06383-f001:**
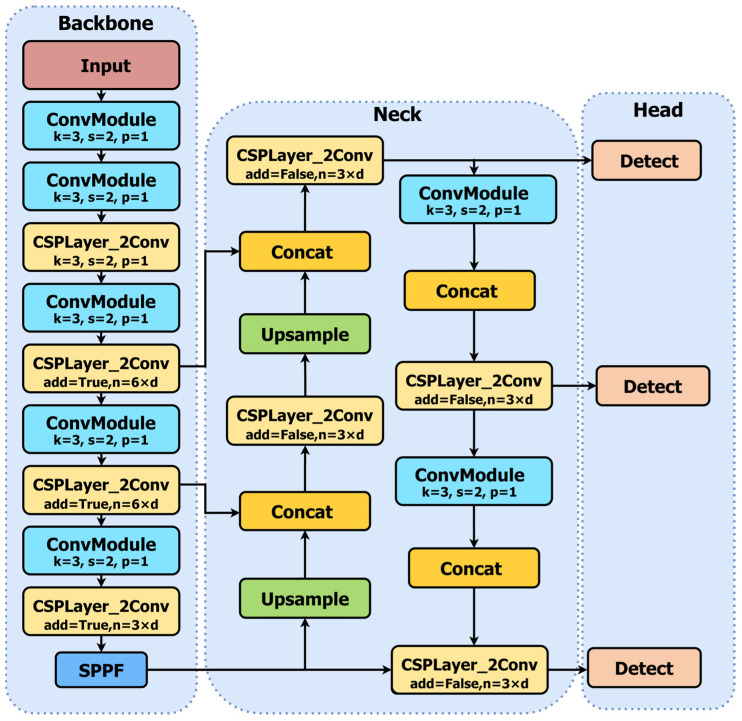
Structure of YOLOv8.

**Figure 2 sensors-24-06383-f002:**
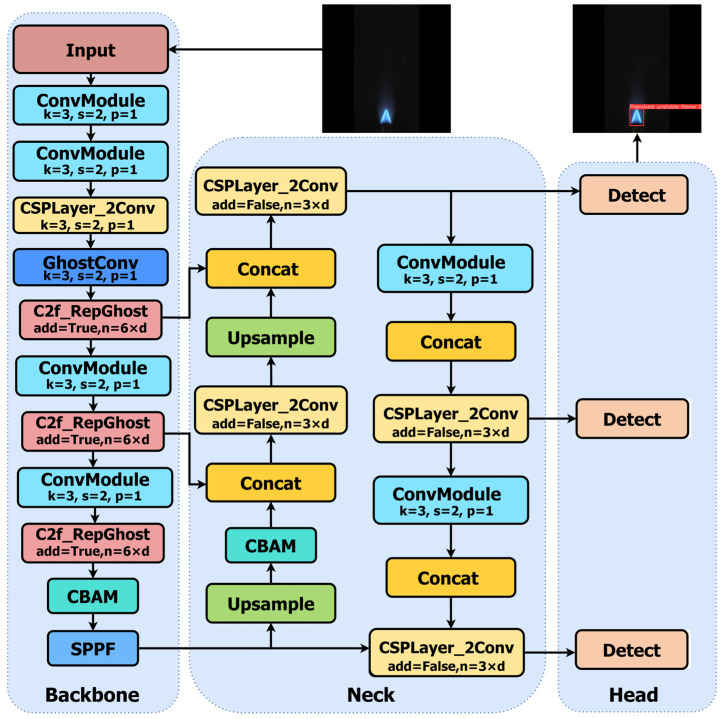
Improved structure of YOLOv8.

**Figure 3 sensors-24-06383-f003:**
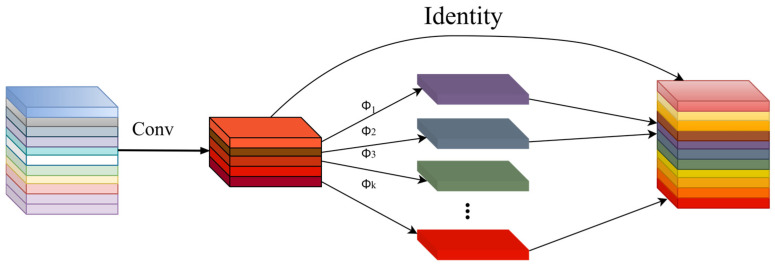
GhostConv convolution module.

**Figure 4 sensors-24-06383-f004:**
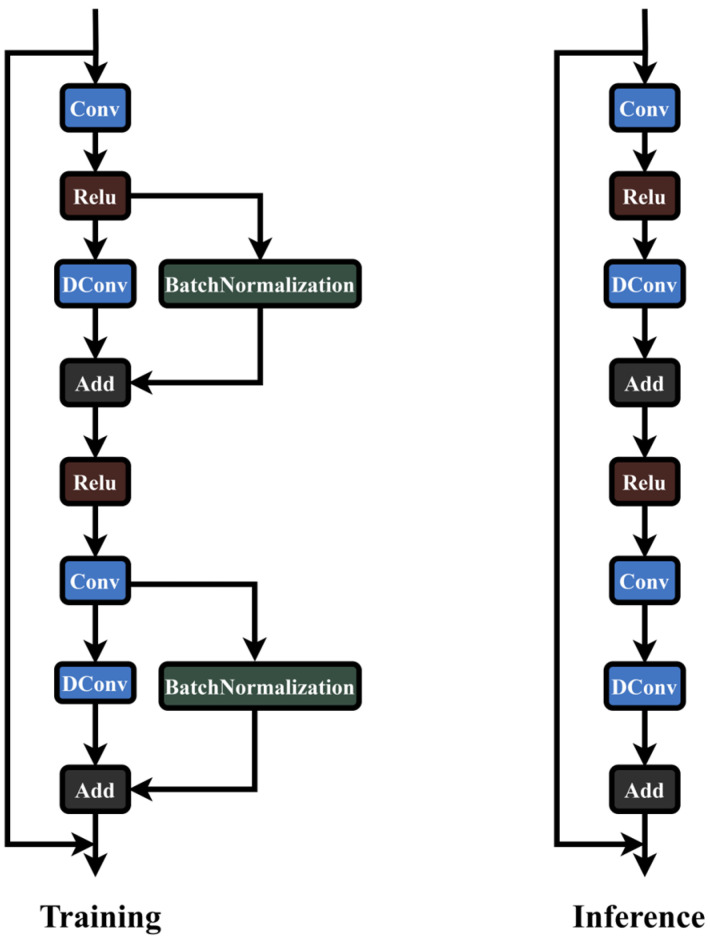
Reparameterization operation of RepGhost.

**Figure 5 sensors-24-06383-f005:**
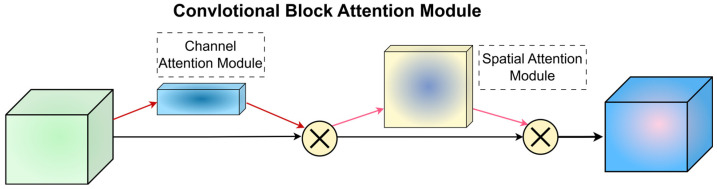
Structure of CBAM.

**Figure 6 sensors-24-06383-f006:**
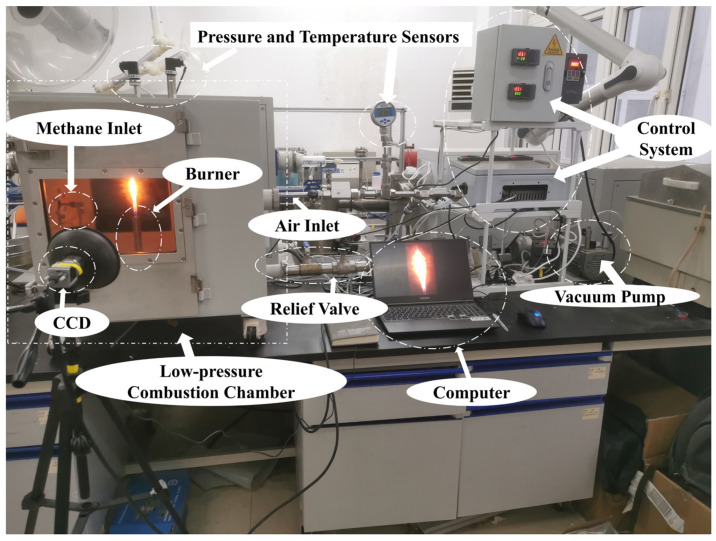
Experimental equipment.

**Figure 7 sensors-24-06383-f007:**
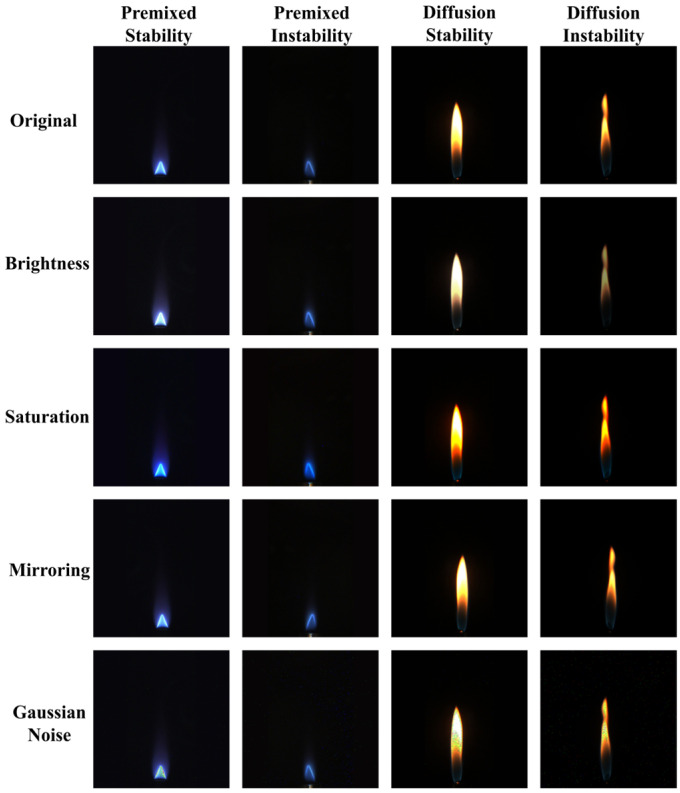
Data augmentation of flame images.

**Figure 8 sensors-24-06383-f008:**
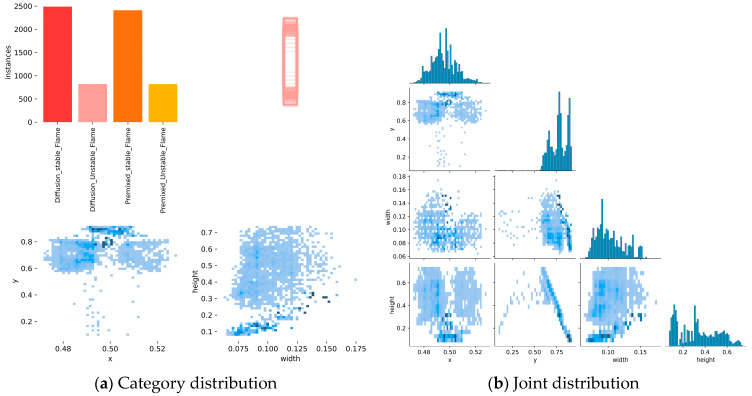
Visualization of dataset distribution.

**Figure 9 sensors-24-06383-f009:**
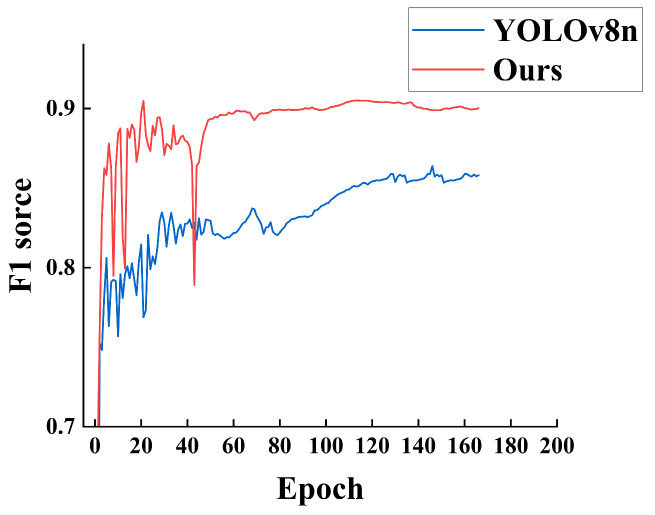
Model F1 score.

**Figure 10 sensors-24-06383-f010:**
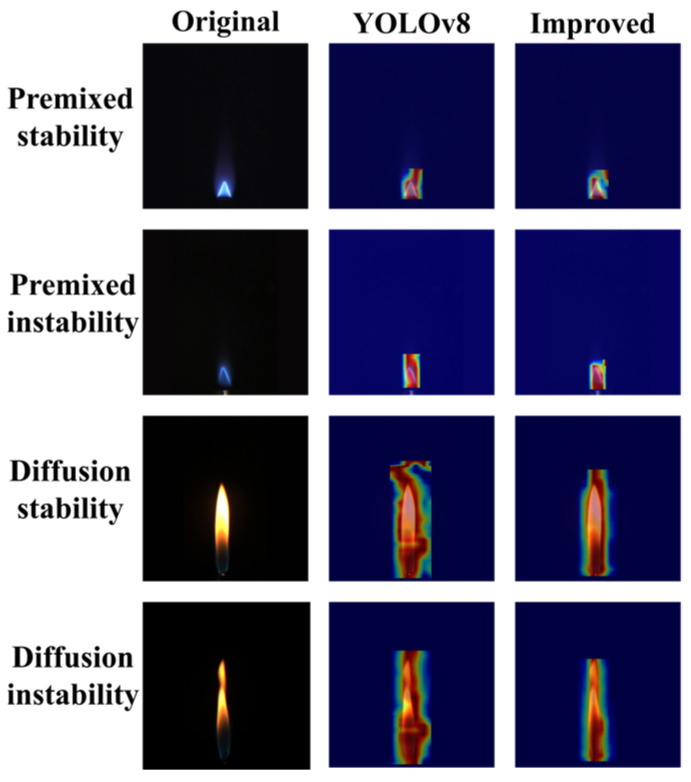
Grad-CAM visualization of flame heatmap.

**Figure 11 sensors-24-06383-f011:**
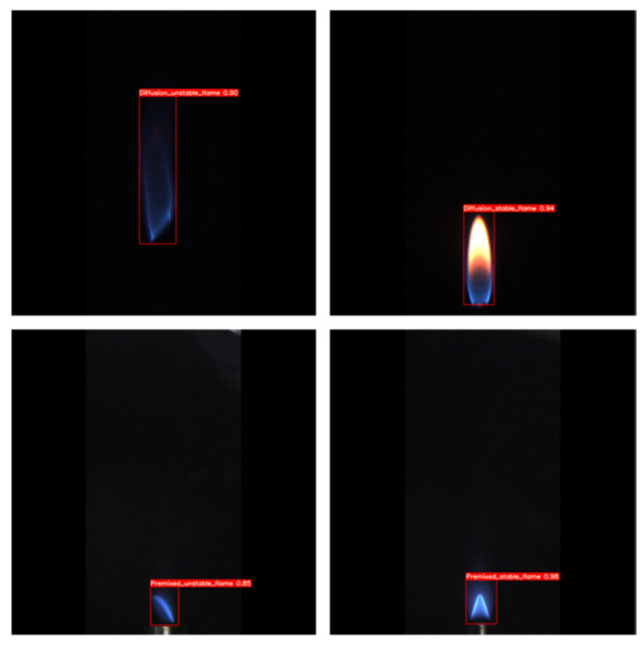
Model prediction results.

**Figure 12 sensors-24-06383-f012:**
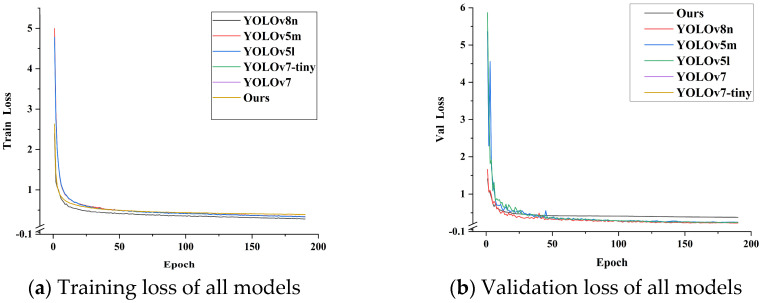
The training curves of all models.

**Figure 13 sensors-24-06383-f013:**
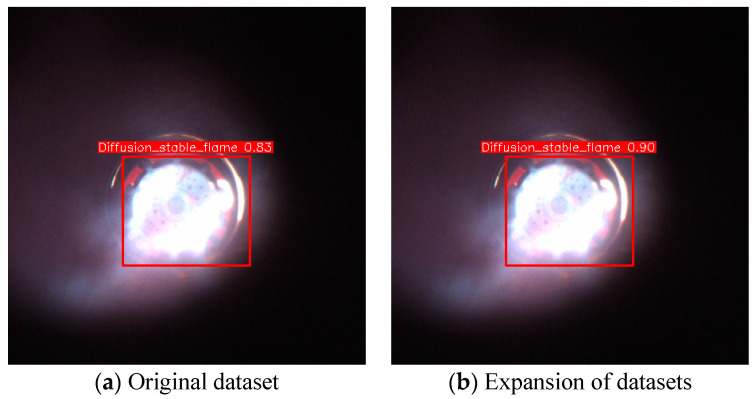
Validation effect of actual flame data.

**Figure 14 sensors-24-06383-f014:**
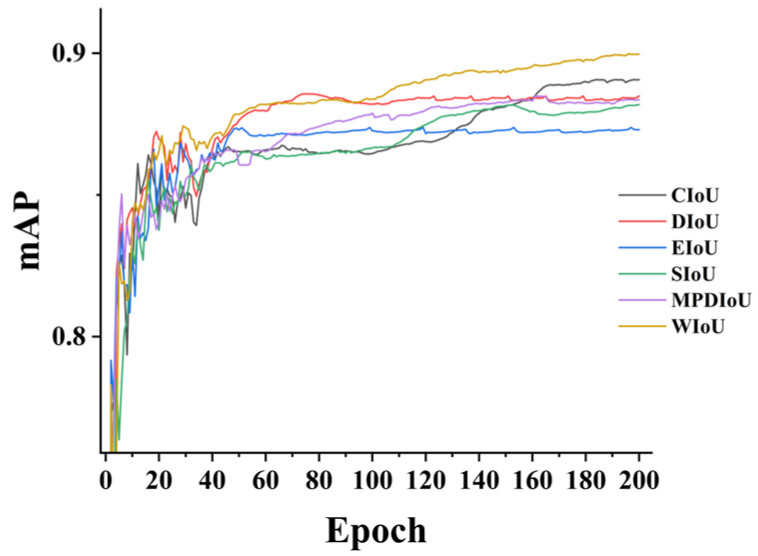
Comparison of the map curves.

**Figure 15 sensors-24-06383-f015:**
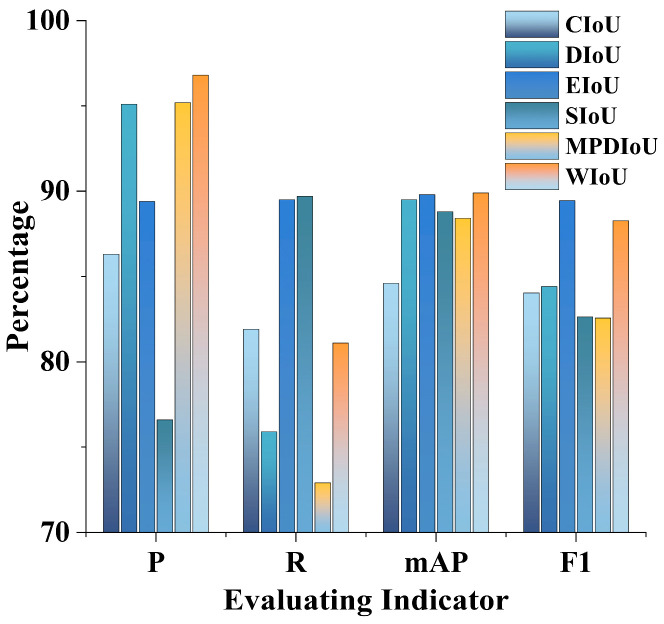
Comparison of other metrics.

**Table 1 sensors-24-06383-t001:** Dataset composition.

Flame Type	Training Set/Number	Validation Set/Number
Premixed Stable Flame	2404	601
Premixed Unstable Flame	812	203
Diffusion Stable Flame	2480	620
Diffusion Unstable Flame	816	204

**Table 2 sensors-24-06383-t002:** Comparison of the improved model with the YOLOv8n model.

Model	YOLOv8n	Ours
Params/M	11.47	10.02
FLOPs/G	8.2	7.2
P/%	75.6	96.8
R/%	88.5	81.1
F1/%	81.7	88.3
mAP@0.5/%	89.4	89.9
FPS	91	87

**Table 3 sensors-24-06383-t003:** Comparative results of YOLO-series models.

Model	Params/M	FLOPs/G	P/%	R/%	F1/%	mAP@0.5/%	FPS
YOLOv5m	79.59	47.9	93	76.9	89.15	85.6	42
YOLOv5l	175.95	107.7	88.3	70.7	78.5	85.3	75
YOLOv7	141.96	105.2	88.4	90	89.19	95.5	31
YOLOv7-tiny	22.98	13.2	86.3	90.6	88.4	95.1	43
YOLOv8s	11.83	8.2	87.7	91.2	89.4	88.8	70
Ours	10.02	7.2	96.8	81.1	88.3	89.9	87

**Table 4 sensors-24-06383-t004:** Comparison of the results of ablation experiments.

Model	Params/M	FLOPs/G	P/%	R/%	F1/%	mAP@0.5/%	FPS
YOLOv8n	11.47	8.2	75.6	88.5	81.7	89.4	91
+M1	11.43	8.0	95.7	81.2	87.9	89.1	85
+M2	10.15	7.2	74.4	91.4	82.8	82.03	81
+M3	10.1	7.2	86.3	81.9	84.03	84.6	78
+M4	10.02	7.2	96.8	81.1	88.3	89.9	87

## Data Availability

Data are contained within the article.

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
