# Peer review of "Improved YOLOv8 for Gas-Flame State Recognition under Low-Pressure Conditions"

_sensors, 2024, doi:10.3390/s24196383_

Round 1
Reviewer 1 Report
Comments and Suggestions for Authors
This paper presents an enhanced YOLOv8n model designed to efficiently and accurately recognize gas flame states under low-pressure conditions. By integrating the GhostConv module to reduce computational parameters, employing the C2f_RepGhost module to deepen convolution and extend feature dimensions, and adding the CBAM attention mechanism to enhance the model's ability to capture fine-grained features of flames, significant performance improvements were achieved. Experimental results indicate that the proposed model, while meeting real-time detection requirements, not only improves detection accuracy and recall but also reduces the number of model parameters and floating-point operations, making it more suitable for deployment on mobile devices, particularly in the context of low-pressure gas flame detection at high altitudes.
1. The paper discusses the impact of low-pressure environments on methane combustion efficiency and flame stability but does not provide details on the specific challenges this presents for flame detection. Could you please elaborate on the particular difficulties associated with flame detection in low-pressure environments?
2. How does the CBAM attention mechanism enhance the model's ability to capture fine-grained features of flames in both channel and spatial dimensions? Could additional experimental data be provided to support this conclusion?
3. In what specific aspects does the WIoU demonstrate advantages in handling highly overlapping or intersecting bounding boxes? Could additional examples or experimental results be provided for illustration?
4. After replacing CIoU with WIoU, how does this specifically enhance the sensitivity and accuracy of the model's regression loss? Is this improvement universally applicable?
5. The experiments were conducted using a simulated dataset. Please provide a detailed description of the methods used to construct this dataset and its relevance to practical applications. How can we ensure the consistency of the model's performance across different pressure environments?
6. According to the results presented in Table 1, there were improvements in the number of parameters, computational load, and accuracy; however, there was a slight decrease in frame rate. Is it possible to further optimize the model structure or training process to enhance detection speed without compromising accuracy?
7. In the data augmentation process, how are the random adjustments of image brightness, saturation, mirroring, and the addition of random Gaussian noise specifically implemented? Has the impact of these operations on the diversity of the dataset and the model's generalization ability been thoroughly evaluated?
Comments on the Quality of English Language
Can be improved.
Author Response
Hello dear reviewer:
As my response contains images, I have organised the content of my response into a word document for your perusal.
I have taken great care to address each of the issues raised and have made the necessary changes to the manuscript.
Your valuable feedback has been instrumental in enhancing the quality and clarity of our work.
Thank you very much for reviewing the manuscript.

Reviewer 2 Report
Comments and Suggestions for Authors
Dear authors and editors, I will use the MDPI reviewer's suggestions format to evaluate and review this research work, as observed below. Please, carefully respond to each of the following comments that can be found within each question.
1. What is the main question addressed by the research?
The authors propose a lightweight flame detection based on an improved YOLOv8 algorithm. The proposed method achieves high accuracy and fast inference time of gas flame state recognition in a low-pressure environment. The authors used GhostNet in the Backbone to form a GhostConv module that help to reduce inference times, as well as a modification of the C2f module by incorporating RepGhost to perform deep convolutions. Additionally, a CBAM attention mechanism was added to improve the model’s ability to detect flame features. The authors demonstrates to achieve high accuracy results up to 87FPS with 89.9mAP.
2. What parts do you consider original or relevant for the field? What specific gap in the field does the paper address?
The application of gas flame state recognition in a low-pressure environment is interesting. In addition, the use of GhostNet and RepGhost and the attention mechanism is very interesting for the YOLOv8 approach.
3. What does it add to the subject area compared with other published material?
The use of YOLOv8 has barely used in other works related to flame detection. Moreover, the use of GhostConv, C2f_RepGhost, and the attention mechanism CBAM is an important contribution of this work compared to other works.
4. What specific improvements should the authors consider regarding the methodology? What further controls should be considered?
- Please, improve the introduction and state of the art. This section is small, and it is unclear if more related work can be found in the literature.
- Please, improve the dataset construction section with a table that indicates the training, validation, and testing set. This would help in understanding the characteristics of the dataset and how it relates to the proposed methods and results.
- Please, explain in detail if the authors used a training, validation, and testing set to evaluate overfitting, bias, and variance. This explanation should be carried out for each model, and it is key since the proposed results might work very well due to overfitting problems. This can mean that the proposed approach works well with this dataset, but doesn't work very well with similar data (does not have generalization abilities).
- I strongly recommend improving the discussion section considering the overfitting analysis.
- In Table 1, the authors clearly explain that their algorithm performance is better than the YOLOv8 version. However, in Table 2, it can be seen that YOLOv7-tiny obtained a better mAP of 95.1%. It can be seen an improvement in the FPS, but the author should explain better what is the contribution of the article since in the abstract it is only mentioned that the proposed approach works better in accuracy and inference times than YOLOv8.
5. Please describe how the conclusions are or are not consistent with the evidence and arguments presented. Please also indicate if all main questions posed were addressed and by which specific experiments.
Based on the results presented by the authors, the proposed methods outperform other models such as YOLOv8. However, it is hard to fairly evaluate this article, a better explanation of the dataset distribution is required. Usually, the datasets are divided into training, validation, and testing, which help to provide valuable information about overfitting, bias, and variance, which helps to present a better evaluation, understanding, and limitations of the proposed algorithms.
6. Are the references appropriate?
The references are updated and consider mostly articles from the last 5 years. Moreover, the references match this work topic. I just recommend reviewing this article to better understand and explain the feature extraction within the object detection metaarchitectures.
https://www.sciencedirect.com/science/article/pii/S016816991932232X
https://ieeexplore.ieee.org/abstract/document/6036772?casa_token=bhDg9IMvbSMAAAAA:Bnp2E0KK5J99bBmaUNAEez-Cv482ZZNpo3jXAhXYdm4F1XsTiypS-S12ri1UgDR0yZAavKRcFTCUK-k
7. Please include any additional comments on the tables and figures and the quality of the data.
The quality of the tables and figures is ok.
Round 2
Reviewer 1 Report
Comments and Suggestions for Authors
My comments have been well addressed, thanks.
Reviewer 2 Report
Comments and Suggestions for Authors
The authors have responded to the comments made in the previous review successfully. In particular, improvements have been made in the following points:
- Considering all the recommendations made, the introduction has been improved and revised.
- The proposed architecture, along with the details of the database used for this application have been reviewed, and are better explained.
- The figures and their explanation have improved significantly.
- Fixed writing errors in the article.
- The dataset distribution is better explained.
- The conclusions have been improved, and the article's contribution is explained better.
Based on this, I can say that this paper can be published in its current state.